# Standing Desks in a Grade 4 Classroom over the Full School Year

**DOI:** 10.3390/ijerph16193590

**Published:** 2019-09-25

**Authors:** Sharon Parry, Beatriz IR de Oliveira, Joanne A. McVeigh, Joyln Ee, Angela Jacques, Leon Straker

**Affiliations:** 1School of Physiotherapy and Exercise Science, Curtin University, Kent Street, Bentley, Perth WA 6102, Australia; Beatriz.Oliveira@curtin.edu.au (B.I.d.O.); s.ee3@graduate.curtin.edu.au (J.E.); Angela.Jacques@curtin.edu.au (A.J.); L.Straker@curtin.edu.au (L.S.); 2School of Occupational Therapy, Curtin University, Speech Therapy and Social Work, Kent Street, Bentley, Perth WA 6102, Australia; Joanne.Mcveigh@curtin.edu.au; 3Movement Science Laboratory, School of Physiology, University of Witwatersrand, Johannesburg, Gauteng 2188, South Africa

**Keywords:** sedentary behaviour, standing desks, physical activity, musculoskeletal discomfort, children, school

## Abstract

School-aged children are spending increasingly long periods of time engaged in sedentary activities such as sitting. Recent school-based studies have examined the intervention effects of introducing standing desks into the classroom in the short and medium term. The aim of this repeated-measures crossover design study was to assess the sit-stand behaviour, waking sedentary time and physical activity, and musculoskeletal discomfort at the start and the end of a full school year following the provision of standing desks into a Grade 4 classroom. Accelerometry and musculoskeletal discomfort were measured in both standing and traditional desk conditions at the start and at the end of the school year. At both time points, when students used a standing desk, there was an increase in standing time (17–26 min/school day) and a reduction in sitting time (17–40 min/school day). There was no significant difference in sit-stand behaviour during school hours or sedentary time and physical activity during waking hours between the start and the end of the school year. Students were less likely to report discomfort in the neck and shoulders when using a standing desk and this finding was consistent over the full school year. The beneficial effects of using a standing desk were maintained over the full school year, after the novelty of using a standing desk had worn off.

## 1. Introduction

In the digital age where interaction with technology is integrated into many aspects of daily living, young people today are likely to be increasingly engaging with digital technology now and into the future. The use of computers, tablets, and handheld devices has grown in the last 10 years [1] and associated with sedentary behaviours [2,3]. It is therefore important that young people, as workers in the future, develop habits of interacting with technology that are potentially health-promoting rather than detrimental to health. 

In recent years, there is emerging evidence that children are spending a high proportion of waking hours in sedentary behaviours [4,5], defined here as waking behaviours of low energy expenditure (metabolic equivalents of ≤1.5) such as sitting [6]. There is a growing body of research that has indicated that prolonged and uninterrupted sedentary time leads to poor health outcomes in adults, such as cardiometabolic disorders, obesity, and all-cause mortality and depression [7,8,9,10,11,12]. In addition, it has been found that replacing sedentary time with standing in sedentary workers is likely to have beneficial physiological effects [13] with no detrimental impact on work productivity. However, the intensity or dosage of standing for optimal benefit in adults is still unclear [14,15]. In children, excessive sedentary behaviour has been associated with increased cardiometabolic risk factors, obesity, anxiety, and depressive symptoms [5,16,17,18,19]. It has also been found that prolonged sitting is associated with the increased prevalence of musculoskeletal symptoms in children and adolescents [20,21,22].

As children spend a majority of their waking hours at school, there have been a number of classroom-based interventions to combat prolonged sitting by replacing traditional seated desks with standing desks in classrooms [23,24,25,26,27,28]. This is not only a convenient way of providing alternatives to sitting but also a means to educate children about spending less time sitting and setting up good sit-stand behaviours into adulthood. Current evidence has found that the provision of classroom standing desks reduces sitting time by 24–38 min per day during school hours [23,25,26,27]. The provision of classroom standing desks was also found to reduce the likelihood of reporting musculoskeletal discomfort in the neck, shoulders, elbows, or lower back [23]. However, while the current research has reported favourable changes in sitting and standing time and health outcomes following short-to-medium-term interventions (5 weeks to 6 months), little is known about whether these improvements are maintained over time. If children are able to maintain improved standing time, without any detrimental effects on overall physical activity by the provision of a standing desk over the full school year, students may be able to build up standing tolerance and associate traditional sedentary activities, such as computer use, reading, and writing with standing rather than sitting and thereby build resilience to standing. In addition, there are no reports of the effects on musculoskeletal symptoms/discomfort following long-term exposure to a standing desk in children.

Many of the studies examining the effects of classroom standing desks have only measured sedentary time and physical activity during school hours rather than during all waking hours. By measuring whole-day sedentary time and physical activity, it may be possible to assess whether changes to standing time during school result in compensatory changes in sedentary time and physical activity in non-school time.

The current study is a follow-up trial of a subgroup (one classroom) from a recently published classroom standing desk intervention [23]. The aim of this study is to assess effects on sitting and standing time at school and sedentary time and physical activity for all waking hours, as well as the self-reported presence and intensity of musculoskeletal symptoms in one classroom of Grade 4 boys, following the intermittent use of a standing desk over a full school year. 

## 2. Materials and Methods 

### 2.1. Study Design and Equipment

The study was a repeated-measures within-subjects crossover trial. The school funded the provision of 12 standing desks (AlphaBetter Adjustable-Height Stand-UpDesk, SAFCO Products Australia at a cost of A$750 per desk) for half of the classroom. The standing desks had a “fidget bar” installed which allowed students to swing their non-supporting legs while standing. The classroom teacher divided the class so that half of the students were using the standing desks while the other half of the boys used traditional seated desks. After 21 school days (three rotations of the school’s seven-day timetable), approximately every four weeks, the boys swapped desks so that the students that were using the standing desks went back to using the traditional seated desks and the students that were sitting started using the standing desks. After another 21 school days, the students swapped desks again. This pattern of switching desks about every four weeks continued throughout the school year and was maintained and monitored by the teacher (Figure 1). In the last week of the first two rotations, hip and thigh accelerometry data and musculoskeletal discomfort ratings were collected. Data were collected again in the last two rotations of the school year. Therefore, data were collected for each participant in both standing and traditional seated desk conditions both at the start and at the end of the school year.

### 2.2. Participants

Participants were drawn from a convenience sample of 24 students (all male, 9–10 years) of one Grade 4 class from an independent boy’s school in Perth, Western Australia. For this school, the classroom design for Grade 4 consisted of each child being allocated an individual desk for the duration of the year; however, the children moved between their desk, the classroom mat, other specialized classrooms (for example, drama, music, and art), collaborative learning spaces, and the sports grounds with most activities occurring in 40 minute blocks. All students and parents were invited to attend an information session that provided details of the aims and protocol of the study. The study complied with the Declaration of Helsinki and included the need for parental consent and participant assent to take part in the study. All students were eligible to take part in the study unless there were any pre-existing health issues that meant that the students would be unable to stand for extended time periods or if there was an inability to wear an accelerometer on a belt around the hip/waist region or around the thigh. 

The study was approved by the Human Research Ethics Committee at Curtin University (approval number: RDHS-157-16).

### 2.3. Outcome Measures

#### 2.3.1. Accelerometer Data

Participants were asked to wear two Actigraph GT9X Link accelerometers (Actigraph LLC, Pensacola, FL, USA), one attached to the thigh and the other around the waist, to measure sitting and standing time during school hours and sedentary time, and light, moderate, and vigorous physical activity during all waking hours (including school hours). The data were collected at 30 Hz and converted to counts per 15 s epoch for analysis. Short epoch length is recommended for measuring physical activity in children in order to detect the frequent movements of short duration in children [29].

The thigh-mounted accelerometer was attached by a cotton tubular elastic bandage that the boys were able to pull on and off, over their school uniform during school hours. The thigh accelerometer was worn from the start to the end of the school day, except for water activities. This thigh-worn accelerometer measured the time in sitting and standing during school hours using the Actigraph inclinometer function of the accelerometer. When worn on the thigh, it has been found that the Actigraph accelerometer is a valid and reliable method of estimating sitting and standing postures [30,31]. Non-wear time was assessed by visual inspection of the Actigraph accelerometer data.

The second accelerometer was worn around the waist, fitted onto an elastic belt and worn for 24 h/day for the seven days, except during water activities. The validity of the Actigraph GT9X Link accelerometers to assess physical activity intensities in children has not yet been determined. However, the earlier version of the Actigraph GT9X Link accelerometer, the GT3X+ (Actigraph LLC, Pensacola, FL, USA) which utilises the same software as the Actigraph GT9X Link, has been found to be valid in assessing the physical activity intensities in children and youth [32,33]. Waking wear time was assessed by visual inspection of the Actigraph accelerometer data by a trained examiner and also by a customised algorithm (SAS Version 9.3, SAS Institute, Cary, NC, USA) [34]. Waking wear time data were processed using the Actilife version 6 (Actilife software; Pensacola, FL, USA) using the accelerometer cut-points for children developed by Evenson [35] to assess sedentary time (<100 counts per minute (cpm)), and light (101 - < 2296 cpm), moderate (2296 - < 4012 cpm), and vigorous (≥4012 cpm) physical activity. For data to be included in the study, accelerometer data were required from at least one day with a minimum wear time of ten hours [36].

#### 2.3.2. Musculoskeletal Data

Musculoskeletal discomfort was assessed twice daily during the assessment weeks using a modified version of the Nordic Musculoskeletal Questionnaire [37]. The questionnaire consists of a body map, with nine labelled body regions (neck, shoulders, elbows, wrists/hands, upper back, lower back, hip/thighs, knees, and ankle/feet). Participants were required to rate discomfort for each body regions using a numerical rating scale from zero to ten [38]. The questionnaire was completed by the students using a personal laptop computer at the start and the end of the day on most testing days as directed by the classroom teacher. The Nordic Musculoskeletal Questionnaire has been used widely in children to assess musculoskeletal pain or discomfort [39,40] and has good reliability (r = 0.57–1.00) and criterion validity (Cohen’s kappa = 0.76) [41].

### 2.4. Analysis

Linear mixed models were used to assess the differences in sitting and standing time during school hours and the percentage of waking hours spent sedentary and participating in various intensities (light, moderate, or vigorous) of physical activity, across the two time points (start of the school year/end of the school year) and across the two desk conditions (standing desk/traditional seated desk). These models included all valid observed data and were adjusted for the number of minutes of accelerometer wear time.

For the musculoskeletal data, binarised pain scores, “presence of discomfort” (pain rating greater than 0), “absence of discomfort” (pain rating of 0) and pain intensity scores (0–10) were compared between the conditions (standing desk/traditional seated desk) and over the two time points (start of the school year/end of the school year) using mixed effects logistic regression and negative binomial regression models, respectively, both with random subject effects. Condition effects and interaction effects between conditions (standing desk/traditional seated desk) and time points were examined. Results were summarised using pain probabilities and corresponding 95% confidence intervals (CI) and estimated mean pain. All *p*-values were two-sided and *p*-values of <0.05 were considered statistically significant. Data were analysed using STATA/IC v16.0 (StataCorp, College Station, TX, USA).

## 3. Results

One Grade 4 class of 24 male students aged 10–11 years were asked to participate in the study throughout 2017. One student did not consent to take part in the study but was still allocated a standing desk periodically according to the research schedule. In total, 23 students participated in the study over the full school year. None of students reported conditions that made them ineligible to take part in the study.

### 3.1. School-Time Standing and Sitting Time

The students wore the thigh accelerometers for six hours a day during school hours (9.00 a.m. to 3.00 p.m.) At the start of the year, there were complete thigh accelerometer data for 22 students using the traditional desks and 16 students using the standing desks; at the end of the school year, there were complete thigh accelerometer data for 20 students using the standing desks and 20 students using the standing desks.

When students used a standing desk, standing time was significantly greater both at the start of the school year (17 min/school day) and at the end of the school year (26 min/school day). Similarly, sitting time was significantly less at the start of the school year (17 min/school day) and at the end of the school year (40 min/school day). There was no statistically significant difference between standing and sitting time when using a standing or traditional seated desk over time (start of school year/end of the school year), but there was a tendency for standing time when using a standing desk to increase over the year and for sitting time to decrease when using a standing desk over the year (interaction: *p* = 0.062, 95% CI: −66 to 2 min) (Figure 2; Appendix A).

### 3.2. Waking-Time Physical Activity and Sedentary Time

At the start of the year, there were complete waist accelerometer data for 15 students using the traditional desks and 13 students using the standing desks; at the end of the school year, there were complete waist accelerometer data for 17 students using the standing desks and 14 students using the standing desks. The mean number of valid days (SD) at the start of the year was 2.5 (1.7) and at the end of the day it was 3.6 (2.0).

The percentages of overall waking sedentary time and light, moderate, and vigorous physical activity were not significantly different between the standing and traditional desk conditions. Over the full school year, the percentages of waking sedentary time and physical activity did not significantly change from the start to the end of the school year (Table 1).

### 3.3. Muculoskeletal Discomfort

The logistical regression analysis of the dichotomous data indicated that there was a reduced likelihood of participants reporting musculoskeletal discomfort in the neck (*p* = 0.004) and shoulders (*p* < 0.001) when using a standing desk, compared to that when using a traditional seated desk. There was no difference in the likelihood of reporting musculoskeletal symptoms at the start and the end of the school year (Table 2).

There were small statistically significant reductions in the intensity of discomfort reported in the neck, shoulder and knees (0.31–0.44/10) when using the standing desk compared to those when using the traditional seated desk (Table 3). Prior research on adults has suggested intensity differences greater than 0.9/10 may be important in non-clinical occupational populations [42,43,44]; however, given the potential differences in vulnerability and resilience of children, the minimum important difference is not known. There was no difference in the intensity of discomfort for any body region between the desk conditions over time (start of school year/end of school year) (Table 3).

## 4. Discussion

This study examined the standing and sitting time during school hours of students that were provided with a standing desk for blocks of about four weeks, periodically over a full school year. The findings were consistent with other similar school-based studies [23,24,26,27,28] that the provision of a standing desk increases standing time (17–26 min/school day) and reduces sitting time (17–40 min/school day). Previous studies have looked at intervention effects over the short to medium term (five weeks to six months) while this current study followed up with students that were intermittently using a standing desk over a full school year, after the novelty of using a new desk was likely to have worn off. It was found that beneficial sit-stand behaviour developed at the start of the year was not only maintained over the full school year but continued to improve by the end of the year. It may be due to the fact that students were able to adapt to the new sit-stand behaviour which could have positive health benefits and set up good habits into adulthood.

There was no real difference in overall waking sedentary time or physical activity between the start and the end of the year. It did not appear that the use of the standing desk resulted in fatigue or a compensatory reduction in out-of-school moderate or vigorous physical activity. However, there may have been some compensatory increase in sedentary time out of school. It was anticipated that the increased standing time during school hours from using a standing desk would result in a subsequent increase in overall waking physical activity but this did not occur. It is possible that, as the accelerometer measuring the data for all waking time physical activity was mounted at the hip, some of the standing still data could have been misclassified as sedentary time rather than low-intensity physical activity.

There have been inconsistent findings in relation to the impact of standing desks on musculoskeletal symptoms from school-based interventions with some studies reporting that the provision of a standing desk increases neck and back discomfort [45] while others reporting no change in musculoskeletal discomfort [39]. This study found that there was a small reduction in the intensity of the reported neck, shoulder, and knee discomfort when using a standing desk. However, while there was a consistent pattern of a lower intensity of musculoskeletal discomfort reported when using a standing desk, the magnitude of the reduction was small and unlikely to be clinically meaningful. There was no difference in the intensity of the discomfort in each of the body regions between the start and the end of the school year indicating that the reduction in the intensity of regional discomfort was maintained over the full school year.

Students were also less likely to report neck and shoulder pain when using a standing desk at both the start and the end of the school year but there was not a significant condition–time interaction. In addition, students were not more likely to report discomfort in the lower extremities from using a standing desk at either the start or the end of the school year. One of the barriers to installing standing desks in classrooms is the potential negative impact of lower extremity and low back pain or fatigue [43,46]. There was no evidence in the current study that students reported increased discomfort or fatigue in the lower extremities. It may be the fact that as the students had a varied timetable of activities, typical of Grade 4, that the variety in postures and activities meant that the students were not standing for long periods that could lead to discomfort or fatigue. Studies examining an older cohort of students with less variety of classroom activities may find different results. 

A strength of the study was that it was conducted over a full school year, allowing any novelty effect of introducing the "new” standing desk to have diminished. The sample size was small, with data collected from only one classroom from a single school. It is challenging to conduct school-based research that involves substantial costs to the school, in this case equipment purchase. Another limitation of the study was the inability to blind the participants to condition allocation, given the obvious nature of the desk. The optimal study design in this situation was a repeated-measures within-subjects crossover design to allow for equitable access of the students to the standing desks and allow students to act as their own control. The absence of blinding could have influenced the participants self-report of discomfort; however, the year-long duration minimised the impact of this. Further, the objective measurement of sitting and standing behaviour did not rely on participant perceptions. It is anticipated that the results of this study will inform future larger-scale school-based research where there is a potential for the provision of standing desks in schools to positively impact the sit-stand behaviour over a full school year.

## 5. Conclusions

This study appears to be one of the first studies that have examined the intervention effects of providing standing desks in a classroom over a full school year. This study found that increased standing time and reduced sitting time during school hours from periodic use of standing desks were maintained over a full school year. There was no compensatory loss to waking time or out-of-school moderate or vigorous physical activity at the start or at the end of the school year. When using a standing desk, students reported a small reduction in the intensity of neck, shoulder, and knee discomfort and were less likely to report neck or shoulder pain. There was no difference in reported musculoskeletal symptoms between the start and the end of the school year. This as a small study from one class in a single school but there were positive results over the full school year. Future larger-scale studies with a variety of year groups, genders, and settings are needed to explore if these results can translate to different populations.

## Figures and Tables

**Figure 1 ijerph-16-03590-f001:**
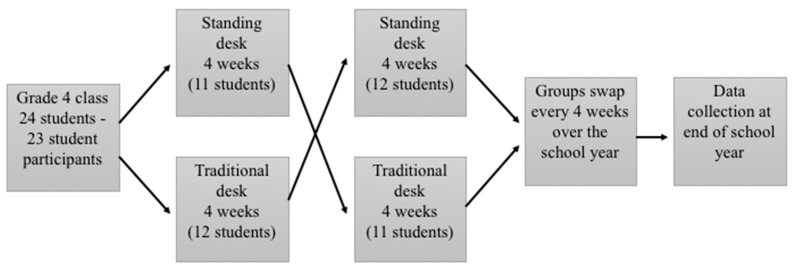
Study design and participant flow.

**Figure 2 ijerph-16-03590-f002:**
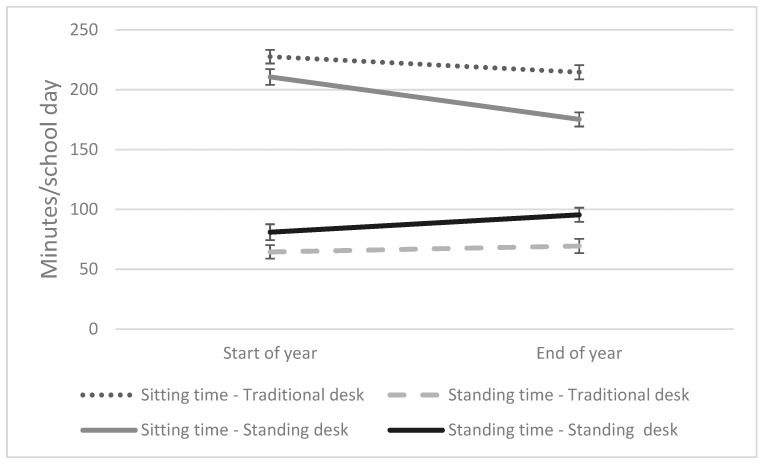
Sitting and standing time at the start and the end of the school year when using a standing desk and a traditional seated desk (min/school day (SE)).

**Table 1 ijerph-16-03590-t001:** Comparison of mean (SD) percentage of sedentary time and light, moderate, and vigorous physical activity between the standing desk and traditional seated desk conditions and over time (start of the school year and end of the school year).

		Start of School Year	End of School Year	Interaction Effect * (95% Confidence Intervals)
Mean (SD) total waking time/day (min)	Standing Desk	785 (92)	759 (92)	
	Traditional Seated desk	770 (119)	730 (66)	
Sedentary time (% waking hours (SD))	Standing Desk	51.4 (2.0)	49.2 (1.9)	
	Traditional Seated desk	48.6 (1.9)	45 (1.8)	−4.6 to 7.3
Light activity (% waking hours (SD))	Standing Desk	42.8 (1.8)	44.6 (1.7)	
	Traditional Seated desk	44.9 (1.7)	48.8 (1.6)	−7.4 to 3.7
Moderate activity (% waking hours (SD))	Standing Desk	4.8 (0.4)	4.9 (0.4)	
	Traditional Seated desk	5.2 (0.4)	5.3 (0.4)	−1.3 to 1.4
Vigorous activity (% waking hours (SD))	Standing Desk	1.1 (0.2)	1.4 (0.2)	
	Traditional Seated desk	1.2 (0.2)	1.0 (0.2)	−0.2 to 1.2

* Interaction between time (start of the school year and end of the school year) and desk type (standing desk or traditional seated desk).

**Table 2 ijerph-16-03590-t002:** Comparison of probability of reporting discomfort in different body regions between standing desk and traditional seated desk conditions and over time (start of school year and end of school year).

		Logistical Regression on Pain Indicator (Probability of Any Pain)					
	Condition	Start of School Year		End of School Year			
		Probability (pain)	95% CI	Probability (pain)	95% CI	*p* *	*p* **
Neck	Traditional Seated Desk	0.50	0.35–0.66	0.50	0.34–0.65	0.890	0.004
	Standing Desk	0.41	0.27–0.55	0.41	0.27–0.55		
Shoulder	Traditional Seated Desk	0.48	0.30–0.66	0.47	0.29–0.65	0.837	<0.001
	Standing Desk	0.37	0.24–0.50	0.37	0.24–0.50		
Elbow	Traditional Seated Desk	0.37	0.24–0.50	0.37	0.24–0.49	0.828	0.489
	Standing Desk	0.35	0.23–0.47	0.35	0.23–0.48		
Wrist	Traditional Seated Desk	0.45	0.29–0.62	0.44	0.28–0.61	0.835	0.479
	Standing Desk	0.43	0.27–0.59	0.43	0.27–0.59		
Upper back	Traditional Seated Desk	0.44	0.29–0.59	0.44	0.29–0.58	0.881	0.640
	Standing Desk	0.43	0.28–0.57	0.43	0.28–0.57		
Lower back	Traditional Seated Desk	0.34	0.23–0.45	0.33	0.23–0.43	0.829	0.327
	Standing Desk	0.32	0.24–0.41	0.32	0.24–0.41		
Hips/thighs	Traditional Seated Desk	0.49	0.33–0.64	0.48	0.32–0.63	0.856	0.985
	Standing Desk	0.49	0.33–0.64	0.49	0.33–0.64		
Knees	Traditional Seated Desk	0.47	0.29–0.65	0.46	0.28–0.64	0.753	0.174
	Standing Desk	0.43	0.26–0.59	0.43	0.26–0.60		
Ankles/feet	Traditional Seated Desk	0.63	0.49–0.76	0.62	0.48–0.76	0.837	0.345
	Standing Desk	0.65	0.54–0.76	0.65	0.53–0.76		

* Interaction effect—difference between standing desk and traditional desk conditions over time (start of school year and end of school year); ** overall difference between traditional seated desk and standing desk conditions.

**Table 3 ijerph-16-03590-t003:** Comparison of pain intensity in different body regions between standing desk and traditional seated desk conditions and over time (start of school year and end of school year).

		Pain Intensity Score							
	Condition	Start of School Year					End of School Year		
		Estimate Mean (Pain Score/10)	Estimate Mean (Pain Score/10)	Estimate Mean Diff	95% CI	*p* *	Overall Estimate Mean Difference Between Conditions	95% CI	*p* **
Neck	Traditional Seated Desk	4.73	4.68	−0.01	−0.21, 0.18	0.943	−0.44	−0.59, −0.29	<0.001
	Standing Desk	3.01	3.01	0.00	−0.23, 0.23				
Shoulder	Traditional Seated Desk	5.06	4.97	−0.02	−0.23, 0.20	0.914	−0.44	−0.61, −0.27	<0.001
	Standing Desk	3.23	3.23	0.00	−0.25, 0.25				
Elbow	Traditional Seated Desk	2.79	2.74	−0.02	−0.28, 0.25	0.896	−0.27	−0.47, −0.06	0.054
	Standing Desk	2.11	2.13	0.01	−0.28, 0.29				
Wrist	Traditional Seated Desk	4.85	4.77	−0.02	−0.20, 0.17	0.908	−0.15	−0.29, −0.01	0.120
	Standing Desk	4.15	4.15	0.00	−0.20, 0.20				
Upper Back	Traditional Seated Desk	3.75	3.69	−0.02	−0.25, 0.21	0.924	−0.09	−0.26, 0.09	0.436
	Standing Desk	3.41	3.41	0.00	−0.25, 0.24				
Lower Back	Traditional Seated Desk	1.61	1.61	0.00	−0.21, 0.21	0.914	−0.17	−0.31, −0.05	0.051
	Standing Desk	1.32	1.31	−0.17	−0.35, 0.02				
Hips/thighs	Traditional Seated Desk	3.34	3.29	−0.02	−0.20, 0.17	0.916	−0.17	−0.31, −0.03	0.084
	Standing Desk	2.80	2.79	0.00	−0.21, 0.21				
Knees	Traditional Seated Desk	4.81	4.72	−0.02	−0.22, 0.18	0.882	−0.31	−0.47, −0.15	0.004
	Standing Desk	3.48	3.50	0.00	−0.22, 0.23				
Ankle/feet	Traditional Seated Desk	5.32	5.24	−0.02	−0.20, 0.17	0.913	−0.10	−0.24, 0.04	0.279
	Standing Desk	4.78	4.77	−0.09	−0.29, 0.10				

* Interaction effect—difference between standing desk and traditional desk conditions over time (start of school year and end of school year); ** overall difference between traditional seated desk and standing desk conditions.

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
