# Peer review of "Standing Desks in a Grade 4 Classroom over the Full School Year"

_ijerph, 2019, doi:10.3390/ijerph16193590_

Round 1
Reviewer 1 Report
The study evaluates the conditions of the students in a classroom in which it was allowed to alternate the sitting position with the standing one.
In the introduction, the authors refer to some articles on the risks to the health of a sedentary lifestyle. It would have been more appropriate if they had referred not to the sedentary lifestyle but to the occupational one [eg.: Lindberg CM, Srinivasan K, Gilligan B, et al. Occup Environ Med 2018;75:689–695). However, at the moment there is not enough evidence that counteracting a sedentary lifestyle in office work causes health benefits [Dupont F, Léger P-M, Begon M, et al. Occup Environ Med 2019;76:281–294. --- Straker L. Occup Environ Med 2019;76:279–280.].
It is not clear how the sedentary lifestyle of adults can be influenced by the fact that the children of the primary schools are allowed to stand up. The authors insist on this point in the conclusions but should explain what mechanism they postulate. In light of the current knowledge, in fact, the hypothesis on which the study is based seems to have little strength.
A consideration, closely linked to the first, concerns the time that the 4th year school students are forced to spend sitting. I have no direct knowledge of the Australian education system and therefore I do not know how to judge (as I believe readers have difficulty doing) if the experimental mechanism is appropriate. In many European schools, in fact, nowadays children spend sitting only part of the time because in the modern classrooms numerous activities take place that requires movement. The classrooms in which they sat for a long time in the pews are a memory of the 50s and 60s of the last century. Authors should provide data on the actual condition of pupils in a normal Australian classroom.
The survey model should be better explained. How many teachers controlled the students, to ensure that actually 12 of them used standing desks and 11 traditional ones, without ever changing places and without getting up, for 21 days? The confusion that reigns in a classroom makes the controls really difficult.
But if the students in the traditional places were really forced to remain stationary for the whole duration of the school timetable, the enthusiasm when they were admitted to the standing desk fully gives a reason for the improvement of musculoskeletal complaints.
The symptoms, in fact, were measured with a subjective questionnaire, initially designed for workers exposed to manual handling of heavy loads, and not very suitable for children. There was no attempt to verify the validity of the Nordic questionnaire responses with a medical examination.
The trial did not take place blindly, because both students and examiners expected the upright posture to be associated with fewer symptoms. This bias severely limits the reliability of the results.
Reviewer 2 Report
Dear authors,
here some suggestions for you.
In Introduction and discussion (implication for research) you should highlight implications of your research for workplaces. Are the children the workers in the future? is it important beginning health promotion even for implications in the workplace? Because the theme of the Special Issue is Occupational behaviour in terms of workplace health prevention and promotion. Here, a citation I suggest to you. Magnavita N, Elovainio M, De Nardis I, Heponiemi T, Bergamaschi A. Environmental discomfort and musculo-skeletal disorders. Occup Med (Lond). 2011 May; 61(3):196-201
In participants selection, I suggest to indicate some demographic details of participants (at least age and sex) and to consider some confounding factors like play sports in extra school time.
I suggest to improve the ethical issue of your paper (cite declaration of Helsinky, explain better aspects related to informed consents and anonymity.
In results show the alpha cronbach of the Nordik Questionnaire.
In results, subparagraph 3.3 when yo say "There was a small significant reduction in the intensity of discomfort", please give some numbers not only on the statistical significance, but also on the dimension of these effects. Therefore, in methods I believe it may be useful to use some further analysis to study the size of the difference in terms of musculoskeletal disorders when using standing desk or traditional seated desk.
In discussion I suggest to say a bit more about the study limitations and strenghts.
Round 2
Reviewer 1 Report
the authors improved the text as far as they could
Reviewer 2 Report
Dear author,
thank you very much for accepting my suggestions. You have just forgot to add a suggested reference. "Magnavita N, Elovainio M, De Nardis I, Heponiemi T, Bergamaschi A. Environmental
discomfort and musculo-skeletal disorders. Occup Med (Lond). 2011 May; 61(3):196-201".
Afterly, I believe that is a better work.